# Alpha-Lipoic Acid Ameliorates Radiation-Induced Salivary Gland Injury by Preserving Parasympathetic Innervation in Rats

**DOI:** 10.3390/ijms21072260

**Published:** 2020-03-25

**Authors:** Jin Hyun Kim, Bae Kwon Jeong, Si Jung Jang, Jeong Won Yun, Myeong Hee Jung, Ki Mun Kang, Tae Gyu Kim, Seung Hoon Woo

**Affiliations:** 1Biomedical Research Institute, Gyeongsang National University Hospital, Jinju 52727, Koreasjjang@gnu.ac.kr (S.J.J.); potato-yun@hanmail.net (J.W.Y.);; 2Institute of Health Science, Gyeongsang National University, Jinju 52727, Korea; blue129j@hanmail.net (B.K.J.); jsk92@gnu.ac.kr (K.M.K.); 3Department of Radiation Oncology, Gyeongsang National University, Jinju 52727, Korea; 4Department of Radiation Oncology, Samsung Changwon Hospital, Sungkyunkwan University School of Medicine, Changwon 51353, Korea; ktg7757@hanmail.net; 5Department of Otorhinolaryngology-Head and Neck Surgery, Dankook University, Cheonan 31116, Korea

**Keywords:** radiation therapy, salivary gland, xerostomia, parasympathetic innervation, Alpha lipoic acid

## Abstract

Radiation therapy is a standard treatment for patients with head and neck cancer. However, radiation exposure to the head and neck induces salivary gland (SG) dysfunction. Alpha lipoic acid (ALA) has been reported to reduce radiation-induced toxicity in normal tissues. In this study, we investigated the effect of ALA on radiation-induced SG dysfunction. Male Sprague–Dawley rats were assigned to the following treatment groups: control, ALA only (100 mg/kg, intraperitoneally), irradiation only, and ALA administration 24 h or 30 min prior to irradiation. The neck area, including SGs, was irradiated evenly at 2 Gy/min (total dose, 18 Gy) using a photon 6 MV linear accelerator. The rats were sacrificed at 2, 6, 8, and 12 weeks after irradiation. Radiation decreased SG weight, saliva secretion, AQP5 expression, parasympathetic innervation (GFRα2 and AchE expression), regeneration potentials (Shh and Ptch expression), salivary trophic factor levels (brain-derived neurotrophic factor and neurturin), and stem cell expression (Sca-1). These features were restored by treatment with ALA. This study demonstrated that ALA can rescue radiation-induced hyposalivation by preserving parasympathetic innervation and regenerative potentials.

## 1. Introduction

Various cancers in the head and neck account for more than 650,000 new cases and 330,000 deaths per year [1]. Radiotherapy is considered as a primary mode of treatment for head and neck cancers, delivered either alone or in combination with surgery and/or chemotherapy [2]. Xerostomia is a radiation-induced side effect resulting from damage to salivary glands (SGs) in head and neck cancers. It leads to a marked deterioration in the quality of life of patients administered radiotherapy, even when the tumor itself has been controlled [3,4,5]. Although studies to protect or restore SG dysfunction by radiation have been performed, most treatments are limited to palliative approaches [6].

Radiation-induced SG dysfunction is characterized by hyposalivation, loss of saliva-producing acinar cells, alterations in the epithelium of the ductal compartment, cell death, reactive oxygen species, inflammation, and fibrosis in SGs [7,8,9,10]. Various cell types exist in SGs such as salivary parenchymal, endothelial, stem/progenitor, and parasympathetic nerve cells. Moreover, damage to SGs resulting from radiation is also caused by the salivary microenvironment, including impaired parasympathetic innervation and regenerative potentials, as well as impairments of the SG itself. It has been reported that rescue or recovery of radiation-induced salivary hypofunction results in the promotion of parasympathetic innervation or functional stem/progenitor cells to allow the regeneration of SGs [11].

Parasympathetic innervation influences organ regeneration. Parasympathetic nerves are a vital component of the progenitor cell niche during development, maintaining a pool of progenitors for organogenesis. Therapeutic irradiation in patients with head and neck cancers lead to reduced parasympathetic innervation as well as SG damage [11]. Functional parasympathetic innervation is associated with functional stem/progenitor cells to protect, replace, or regenerate saliva-producing cells after irradiation.

Alpha lipoic acid (ALA) is a strong antioxidant exhibiting high reactivity toward free radicals, and it elevates glutathione levels in tissues [12]. Recently, we reported that ALA protects against radiation-induced normal tissue injury and dysfunction [7,8,9,10]. However, in our previous studies, saliva is significantly reduced even if the salivary glands were partially intact after irradiation. Therefore, there is a hypothesis that some factor existed to control these salivary glands. Here, we aimed to determine whether restoration of parasympathetic innervation by ALA reduces SG dysfunction after radiation damage.

## 2. Results

### 2.1. Radiation Significantly Induces Body Weight and SG Weight Loss, but ALA Ameliorates Radiation-Induced SG Weight Loss and Reduced Saliva Levels.

Radiation significantly induced body weight loss at all experimental time points. No recovery was observed in the irradiation group after ALA treatment (Figure 1A). Radiation significantly induced the loss of gland wet weight at 6 and 8 weeks after radiation. However, the administration of ALA significantly recovered gland weight at 8 weeks after irradiation compared with that in the irradiated controls (Figure 1B). In addition, radiation significantly reduced saliva secretion, which was partially recovered by ALA (Figure 1C). These data suggest that ALA may be involved in the recovery of irradiation-induced hyposalivation.

### 2.2. ALA Restores Radiation-Induced AQP5 Expression.

The secretion of saliva occurs via aquaporin5 (AQP5) during the process of SG water discharge. Irradiation significantly reduced the expression of AQP5 protein in SGs at 2, 6, 8, and 2 weeks after irradiation, but ALA significantly improved the AQP5 expression level to the same extent each week after irradiation as indicated by Western blot analysis (Figure 2A). However, unchanged expression of AQP5 was observed in the control and ALA-only groups (Appendix A). The preservation of AQP5 protein expression by ALA administration was further confirmed by immunohistochemical staining. Consistently, AQP5-positive acinar cell numbers were significantly decreased in the irradiation group at each time point but were significantly restored by ALA, as shown by Western blotting (Figure 2B). These data indicate that ALA contributed to the recovery of salivary function impaired by irradiation.

### 2.3. ALA Preserves Parasympathetic Innervation

Radiation affects parasympathetic innervation in SGs, and parasympathetic stimulation improves regeneration of SGs after irradiation [13,14]. To examine the effects of ALA on radiation-induced damage of the parasympathetic nerve in SGs, we investigated the protein levels of glial cell-line derived neurotrophic factor family receptor alpha 2 (GFRα2), a marker of parasympathetic innervation, in SGs. The radiation-induced impairment of GFRα2 protein expression was significantly ameliorated by ALA administration (Figure 3A, D). However, expression level of GFRa2 in ALA-only groups was not significant different to control groups (Appendix A). This was confirmed by immunohistochemical staining of acetylcholinesterase (AchE), another marker of parasympathetic innervation (Figure 3B,D), and by immunofluorescent staining of neurofilaments of parasympathetic innervation (Figure 3C,D). These data suggest that ALA protects SGs from radiation-induced SG hypofunction by preventing parasympathetic innervation in SGs.

### 2.4. ALA Improves Neurotrophic Factor Levels in SGs.

The radiation-induced impairment of parasympathetic innervation in SGs is rescued by the increased production of neurotrophic factors [13]. The brain-derived neurotrophic factor (BDNF) and neurturin levels in SGs were significantly reduced by radiation at each time point, but the BDNF level was significantly improved at 2 and 8 weeks, and the neurturin level at 8 and 12 weeks, after irradiation in the ALA-treated irradiation group (Figure 4A). No significant difference was found in the serum BDNF and neurturin level in all groups at all time point. (Figure 4B). These data indicate that ALA preserves parasympathetic innervation by maintaining neurotrophic factor levels in SGs following irradiation.

### 2.5. ALA Increases SG Regeneration

Hedgehog (Hh) signaling is known to be activated during the functional regeneration of adult SGs after duct ligation [15]. Moreover, transient activation of the Sonic Hedgehog (Shh) gene rescues radiation-induced hyposalivation [13]. We investigated the effect of ALA on radiation-induced Hh signaling. The mRNA expression levels of Shh and its receptor Patched (Ptch), Hh signaling and regeneration-related factors, were measured by qPCR. In the irradiation group, Shh and Ptch Mrna levels were decreased; however, in the irradiation group treated with ALA, the mRNA level of Shh was increased at 2 and 6 weeks and that of Ptch at 2, 6, and 8 weeks (Figure 5). This suggests that ALA is involved in SG regeneration after irradiation via Hh signaling.

### 2.6. ALA Rescues the Endogenous Resident Stem Cell Population.

To examine the effect of ALA on the regenerative potential of SGs, we assessed the production of stem cells in irradiated glands treated with ALA. Rodent SG stem cell/progenitor cells express well-established stem cell markers, including Sca-1 [16]. However, the number of Sca-1-positive cells is extremely low in vivo. Thus, we performed reverse-transcription PCR to assess Sca-1 expression. Irradiation alone significantly reduced Sca-1 mRNA expression at 6, 8, and 12 weeks after irradiation, suggesting a deficit in resident SG stem/progenitor cells compared with non-irradiated controls. Rats that received ALA demonstrated significantly enhanced Sca-1 mRNA expression compared with the irradiated group (Figure 6A,C). To confirm effect of ALA on the preservation of stem cell population, we performed immunostaining for c-Kit. c-Kit is also known to SG stem/progenitor cell markers in mice and rodents [17]. c-Kit-positive signals were detected in the excretory ductal cells of the submandibular gland (arrow in Figure 6B). The localization of c-Kit-positive signals coincided well with the previous report [17]. Irradiation significantly downregulated c-Kit expression particularly in ductal structures in all time points (RT), whereas ALA treatment prior to irradiation restored c-Kit expression significantly in 2 weeks, but tended to ameliorate in other time points (ALA + RT) (Figure 6B,D). These data suggest that ALA can rescue resident stem cells in the irradiated SG environment.

## 3. Discussion

The present study showed that ALA can promote salivary regeneration by restoring parasympathetic innervation and preserving resident stem cell populations after irradiation. In the previous study, we demonstrated that the loss of secretory acinar cells was the major cause of xerostomia after irradiation, and that ALA restored salivary acinar cells by reducing apoptosis, inflammation, and fibrosis [10]. However, this effect of ALA may be temporary and is limited to SG tissue itself. The early effects of radiation therapy may be induced by salivary tissue membrane damage, whereas more delayed and long-term effects have been proposed to be a consequence of radiation-induced damage of progenitor cells within the adult SG, leading to a compromise in the regenerative capacity of the gland [18]. Finally, these long-term effects after irradiation are a major cause of xerostomia. Thus, there is a shift to change the focus from SG tissue restoration to extra-SG tissue regeneration. Saliva secretion is closely related to neurotransmitters, secreted from parasympathetic nerves, which binds to and activates its receptor in SG cells [19]. This receptor, when coupled with acetylcholine, activates downstream signaling and consequently elevates the intracellular calcium level in apical salivary cells [14]. Indeed, the anatomy of gland innervation may, in part, explain the effects of radiation therapy on autonomic nerves. Because the cell bodies of parasympathetic ganglia are located within SGs, they may be more affected by radiation therapy, whereas sympathetic nerve cell bodies are located in the superior cervical ganglion, and are thus distant from the site of radiation therapy [19,20,21,22]. The maintenance of a progenitor cell population as a reservoir of undifferentiated cells is required for the development and regeneration. Knox et al. reported that the removal of the parasympathetic ganglion in mouse explant organ culture decreased the number and morphogenesis of keratin 5-positive epithelial progenitor cells [23]. They also demonstrated that acetylcholine signaling, via the muscarinic M1 receptor and epidermal growth factor receptor, increased epithelial morphogenesis and the proliferation of keratin 5-positive progenitor cells. Parasympathetic innervation prevents the epithelial progenitor cell population from differentiating, which is required for organogenesis. Therefore, the mechanism underlying parasympathetic innervation maintenance may be a therapeutic target for organ repair or regeneration. In particular, it has been reported that activation of acetylcholine and its receptor leads to increased levels of cytoplasmic calcium, which plays an important role in the translocation of AQP5, a water transport channel protein, in SG cells [23]. This suggests that water transport in SG is controlled by parasympathetic signal transduction. In this experiment, increased AQP5 expression by ALA may have resulted from protection of parasympathetic innervation by ALA (Figure 2 and Figure 3). The neurotrophic factor neurturin regulates parasympathetic ganglion function, which affects SG development as well as salivation, and interacts with the receptor GFRα2. We found that ALA is involved in enhanced parasympathetic protection via the rescue of GFRα2 and AchE expression and the preservation of the levels of the nerve growth factors BDNF and neurturin in SGs (Figure 3 and Figure 4).

This suggests that ALA ameliorates the radiation-induced impairment of parasympathetic innervation in SGs. In addition, Hh signaling rescues radiation-induced hyposalivation [24]. Transient activation of the Hh pathway by Shh gene delivery may rescue salivary function after irradiation, and the Hh/Gli pathway may primarily function non-cell-autonomously to achieve the rescue effect. We showed that ALA significantly increased the mRNA expression levels of Shh and Ptch mRNA, which are Hh signaling and regeneration-related factors (Figure 5). Previous studies showed that pronounced regenerative potentials are generated by injury in SGs [25,26,27], as indicated by the proliferation and differentiation of ductal and acinar cells and the rather rapid return of saliva secretion to basal levels. Acinar cells of SGs are replaced by the stem cell population [28]. Aure et al. reported that acinar cell proliferation accounts for the postnatal growth and expansion of SGs, as well as for the maintenance and regeneration of the adult organ [29]. In addition, Pringle et al. showed that the low-dose radiation used to induce SG hypofunction permits the survival of endogenous stem/progenitor cells with regenerative potential in a murine model [16]. Stem/progenitor cells exist in rodent SGs and express surface markers, including Sca-1 [16]. According to our data, ALA significantly recovered Sca-1 and c-kit expression reduced by radiation (Figure. 6), suggesting that ALA is involved in the preservation of resident stem cells as well as activation of parasympathetic innervation after irradiation.

Although adult stem cells are advantageous for therapeutic promise in regenerative medicine, translation to the clinic is uncommon due to factors such as few stem cell populations, immune rejection, long-term engraftment capacity, and less functional rescue potential. To overcome this limitation, stem cells should be provided as an endogenous boost of SG function against injury. The incidence of head and neck cancer has increased in younger patients as well as in the elderly. Therefore, it may be necessary to stimulate or enhance regenerative potential using clinically relevant agents that have been verified for clinical use. ALA is now in clinic use and is mainly used for improving symptoms of the patients with diabetic neuropathy. However, its use and effects are still controversial in other clinical conditions. Although this study does not show the effects of ALA on head and neck cancer, we believe that ALA is capable of treating radiation-induced neuropathy. In conclusion, the current data indicate that ALA has a promising therapeutic potential against radiation-induced salivary dysfunction.

## 4. Materials and Methods

### 4.1. Ethics Statement

The Gyeongsang National University Institutional Animal Care and Ethics Committee approved this study (GLA-120120-R0002) on Jan. 20, 2012.

### 4.2. Radiation Exposure

We assigned male Sprague–Dawley rats (230–250 g; Koatech Inc., Peongtaek, Korea) to the following groups: control, *n* = 12 (Con); ALA administration alone, *n* = 12 (ALA); irradiation alone, n = 16 (RT); and ALA administration before irradiation, *n* = 16 (ALA + RT). Each number of animals reflects numbers in all time points (2, 6, 8, and 12 week) per individual experiment. Three experiments were performed independently. We administered ALA (100 mg/kg, intraperitoneally; Bukwang Pharmaceutical Co., Seoul, Korea) either 24 h or 30 min before irradiation, and we chose the dose and frequency based on previous studies. [19,23,24] The neck area was evenly irradiated with 2 Gy/min (total dose, 18 Gy) using a photon 6- MV linear accelerator (21EX,; Varian, Palo Alto, CA, USA). A 3- cm block of Lucite was positioned above the head and neck to provide adequate buildup and facilitate even radiation distribution. Each rat was exposed to a single dose of radiation and was sacrificed 2, 6, 8, or 12 weeks after radiation.

### 4.3. Salivary Gland Function

Salivary functional activity was evaluated through the measurement of saliva secretion. Pilocarpine (1 mg/kg, intraperitoneally, Isopto Carpine; Alcon Korea Ltd., Seoul, Korea) was injected, and after 8 min the saliva output was collected from the floor of the mouth for 5 min. The collected saliva was placed in pre-weighed 1.5 mL tubes, and the volume was normalized to body weight. Salivary lag times and flow rate were also measured. Salivary flow rates (total saliva weight divided by the collection time) and lag time (time from stimulation to the commencement of saliva secretion) were calculated.

### 4.4. Immunoblotting

SGs were homogenized in lysis buffer. The resulting proteins (50 µg) were loaded on a sodium dodecyl sulfate–polyacrylamide gel and electroblotted. The blots were probed with primary antibodies against polyclonal anti-aquaporin 5 (AQP5) (Abcam, Cambridge, MA, USA) and anti-glial cell-derived neurotrophic factor family receptor α2 (GFR α2) (Abcam) at 4 °C overnight. The primary antibody was visualized by a secondary antibody and an enhanced chemiluminescence kit (Amersham Pharmacia Biotech, Piscataway, NJ, USA).

### 4.5. Immunohistochemistry

After deparaffinization, the sections were incubated with primary antibodies against polyclonal anti-AQP5 (Abcam), anti-acetylcholinesterase (Elabscience, Houston, TX, USA), anti-neurofilaments (Abcam, Cambridge, UK), and anti-c-Kit (Sigma, St. Louis, MO, USA) followed by Alexa 488 fluorophore conjugated secondary antibody and biotin-conjugated secondary IgG (diluted 1:200; Vector Laboratories, Burlingame, CA, USA), avidin–biotin–peroxidase complex (ABC Elite Kit; Vector Laboratories, Burlingame, CA, USA), and diaminobenzidine tetrahydrochloride. Next, we visualized the section by light or epifluorescence microscopy, captured, and analyzed the digital images.

### 4.6. Enzyme-Linked Immunosorbent Assay (ELISA)

To verify the effects of ALA on salivary regeneration, ELISAs for neurotrophic factors were performed. Fresh tissues and serum were collected and stored at −80℃. The levels of BDNF (Quantikine ELISA kit; R&D Systems, Minneapolis, MN, USA) and neurturin (ELISA kit; Elabscience, Houston, TX, USA) were measured according to the manufacturer’s instructions.

### 4.7. Quantitative Real-Time Polymerase Chain Reaction (qPCR)

The transcript levels of Shh and Ptch, key proteins involved in Hh signaling, were measured by qPCR. Salivary tissues were resuspended in TRIzol Reagent (Invitrogen Life Technologies, Carlsbad, CA, USA), and total RNA was extracted. Purified RNA was subsequently reverse transcribed into cDNA using the iScript cDNA synthesis kit (Bio-Rad Laboratories, Hercules, CA, USA). After reverse transcription, quantitative cDNA was amplified using the TaqMan gene expression assay mix (Shh; Rn00568129; Ptch: Rn01527980) on the Applied Biosystems qPCR system (Applied Biosystems Inc., Foster City, CA, USA). The thermal cycle conditions were as follows: denaturation at 95°C for 3 min, followed by 50 cycles of denaturation at 95°C for 10 s, and annealing and extension at 60°C for 30 s. Glyceraldehyde 3-phosphate dehydrogenase (GAPDH) was used as an internal control for normalization of RNA quantity. The relative gene expression level in each sample was quantified using the 2^-ΔΔCt^ method.

### 4.8. Reverse-Transcription PCR

Total RNA was extracted from salivary tissues using the TRIzol method (GIBCO BRL, Grand Island, NY, USA). RNA (5 µg) was converted into cDNA, and the resulting cDNA (2.0 μL) was subjected to PCR amplification. The primer sequences were as follows: 5′-AACCATATTTGCCTTCCCGTC-3′ (sense) and 5′-GAGATCTGAAAGCCCTAGAG-3′ (antisense) for Sca-1, and 5′-TCCCTCAAGATTGTCAGCAA-3′ (sense) and 5′-AGATCCACAACGGATACATT-3′ (antisense) for GAPDH.

### 4.9. Statistical Analysis

Statistical analyses were performed using Graph Pad Prism 8 (Graph Pad Software Inc., La Jolla, CA, USA). The Mann–Whitney *U* test was used to examine the differences between two groups. A *P*-value < 0.05 was considered significant.

## 5. Conclusions

ALA ameliorates the hyposalivation of SGs induced by therapeutic irradiation in two ways. First, it protects salivary cells by preserving the signaling induced by parasympathetic innervation. Second, it promotes the proliferation and differentiation of salivary cells by emanating regenerative signals from resident stem/progenitor cells. Therefore, we propose that ALA is a potential agent against radiation-induced hyposalivation in patients with head and neck cancers. 

## Figures and Tables

**Figure 1 ijms-21-02260-f001:**
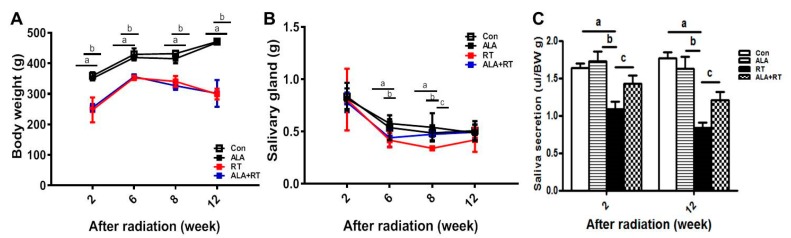
ALA ameliorates the SG weight loss and impaired saliva secretion induced by irradiation. Sprague–Dawley rats were subjected to 18 Gy irradiation in the head and neck region. (**A**) Body weight, (**B**) SG weight, and (**C**) saliva levels were measured at each time points after irradiation. N = 3-4 SGs/group. a, control vs. irradiation group. b, ALA vs. irradiation groups. c, irradiation vs. ALA plus irradiation groups. Values are represented as the mean ± SEM. ^a,b,c^
*p* < 0.05.

**Figure 2 ijms-21-02260-f002:**
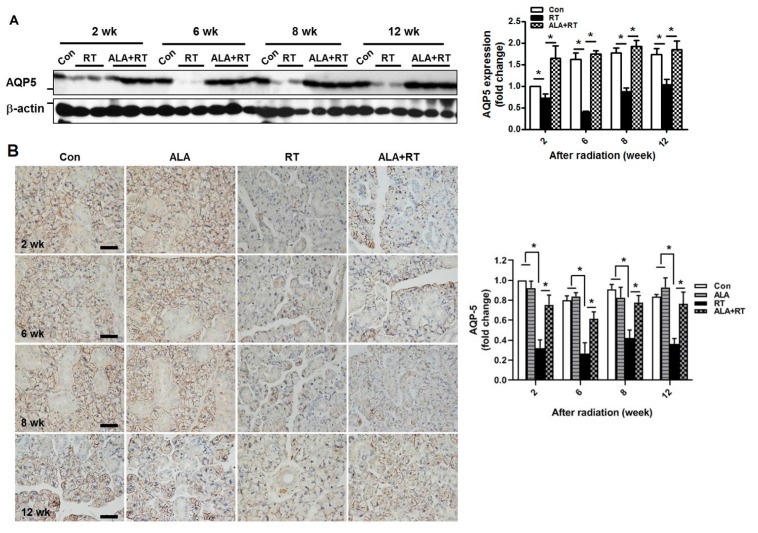
The administration of ALA improves radiation-induced AQP5 expression. Sprague–Dawley rats were subjected to 18 Gy radiation in the head and neck region. (**A**) SGs were collected at each time points after irradiation and subjected to Western blotting. β-actin was used as the loading control. Western blot for the expression of AQP**5** is normalized to β-actin, the expression was shown as signal intensity, and the expression in each group was represented as fold change. The fold change is calculated as the ratio of the final value in each group to the value in the control group at 2 weeks (set as “1”). Size markers mean 25 and 50 kDa from upper. Con; control (*n* = 3). RT; irradiation only group (*n* = 4). ALA+RT; ALA plus irradiation groups (*n* = 3). (**B**) immunohistochemical staining of AQP5 was shown in representative images. Positive signals were calculated as signal density. The fold change is calculated as the ratio of the final value in each group to the value in control group at 2 weeks (set as “1”). Scale bar, 50 μm. *N* = 3-4 SGs/group. Values are represented as the mean ± SEM. **p* < 0.05.

**Figure 3 ijms-21-02260-f003:**
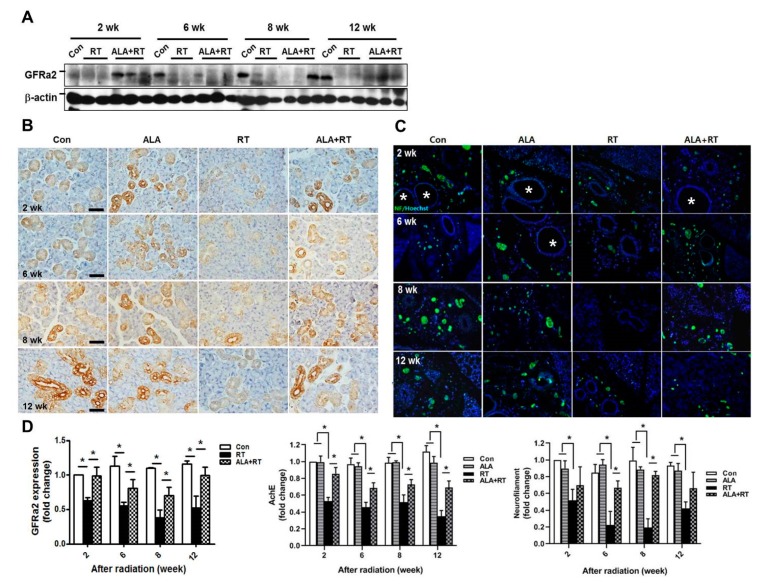
ALA rescues parasympathetic innervation in SGs. Sprague–Dawley rats were subjected to 18 Gy radiation in the head and neck region. (**A**) SGs were collected at each time points after radiation and subjected to Western blotting of GFRα2. β-actin was used as the loading control. The expression of GFRa2 is normalized to β-actin, was shown as signal intensity, and the expression in each group was represented as fold change. The fold change is calculated as the ratio of the final value in each group to the value in control group at 2 weeks (set as “1”). Size markers mean 50 kDa. Con; control (*n* = 3). RT; irradiation only group (*n* = 4). ALA+RT; ALA plus irradiation groups (*n* = 3). The immunohistochemical staining of AchE (brown in (**B)** and neurofilaments (**C**) was shown in representative images. * indicate the ducts in SGs in C. Scale bar, 50 μm. *N* = 3–4 SGs/group. (**D**) Western blot and positive signals for each target were calculated as signal density. Values are represented as the mean ± SEM. **p* < 0.05.

**Figure 4 ijms-21-02260-f004:**
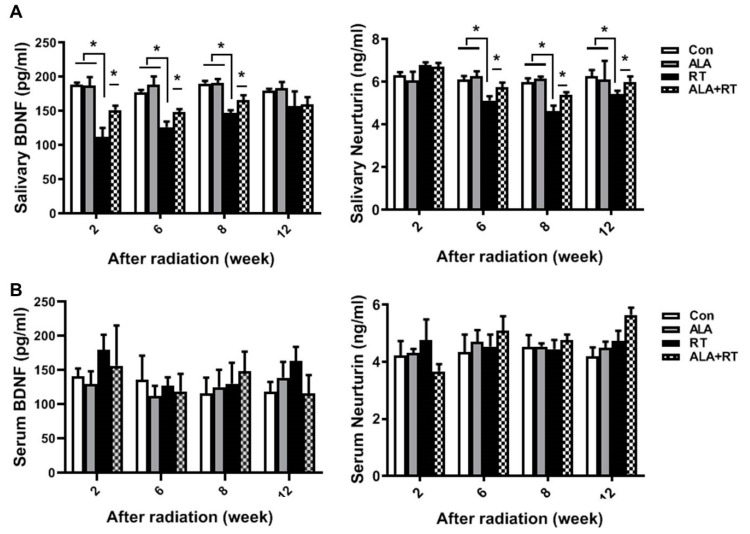
ALA maintains neurotrophic factor levels in SGs. Sprague–Dawley rats were subjected to 18 Gy irradiation in the head and neck region. SGs and serum were collected at each time points after irradiation and subjected to ELISA to measure BDNF (left in A and B) and neurturin (right in A and B) levels. *N* = 12–16 SGs/group. Values are represented as the mean ± SEM. * *p* < 0.05.

**Figure 5 ijms-21-02260-f005:**
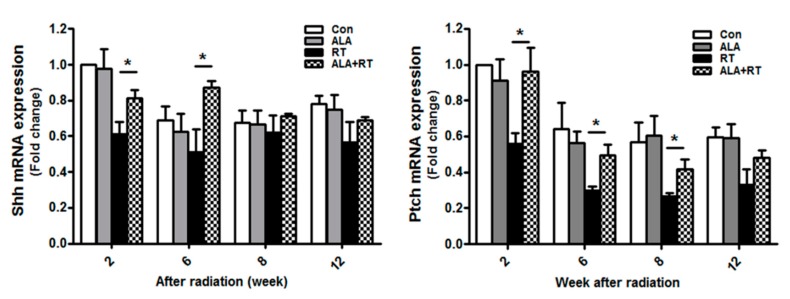
ALA rescues Hh signaling in SGs after irradiation. Sprague–Dawley rats were subjected to 18 Gy radiation in the head and neck region. SGs were collected at each time points, and RNA was extracted and subjected to qPCR analysis of Shh and Ptch expression. *N* = 12–16 SGs/group. Values are represented as the mean ± SEM. **p* < 0.05.

**Figure 6 ijms-21-02260-f006:**
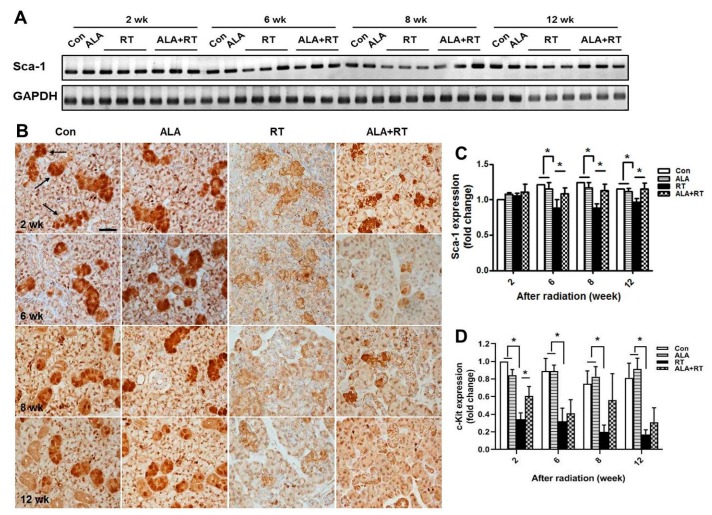
ALA ameliorates radiation-induced impairment of stem cells. Sprague–Dawley rats were subjected to 18 Gy radiation in the head and neck region. (**A** and **C**) SGs were collected at each time points, and RNA was extracted and subjected to reverse-transcription PCR analysis of Sca-1. GAPDH was used as the loading control. The expression level of Sca-1 is normalized to GAPDH. The expression in each group was represented as fold change. Con; control (*n* = 3). ALA; ALA only treated group (*n* = 3). RT; irradiation only group (*n* = 3). ALA+RT; ALA plus irradiation groups (*n* = 3). (**B** and **D**) The sections were immunostained with anti-c-Kit. Figures are representative images from each group. c-Kit positive signals on SG sections (B) and the quantification of staining intensity (**D**). Scale bar, 50 μm. *N* = 3–4 SGs/group. The fold change is calculated as the ratio of the final value in each group to the value in control group at 2 weeks (set as “1”). Values are represented as the mean ± SEM. * *p* < 0.05.

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
