# Peer review of "Alpha-Lipoic Acid Ameliorates Radiation-Induced Salivary Gland Injury by Preserving Parasympathetic Innervation in Rats"

_ijms, 2020, doi:10.3390/ijms21072260_

Round 1

Reviewer 1 Report

ITs difficult to get excited by this manuscript as these authors have already shown the lipoic acid effect on irradiation induced damage of salivary glands before.  Specifically they have also shown the AQP5 and the salivary secretion results before.  What is new and more interesting is the effects on nerves and although the GFRa2, neurturin and BDNF results are interesting what is missing is the functional tests of the nerves.  THis could be achieved in two ways -either electrically stimulating nerves or using taste to show functional integrity.  Their acetyl cholinesterase staining is poor and does not show the nerves.  Thus we are left with an unfinished conclusion that lipoic acid probably preserves the nerves in the same way that the parenchyma is preserved.  

The other results, looking at stem cell markers is too speculative to be useful.  qPCR may show message levels but nothing more and does not show regenerative potential.  I think most of these results should be removed. 

The writing is generally ok but  there are lapses in understanding eg in discussion suggesting that parasympathetic stimulation increases  apical intracellular calclium (which is does) which is then released into the ER and the ductal (which it does not).  

Minor points: 

Hoffman is not a suitable reference

Fig 3 legend makes no sense. Fig 3a needs MW markers, B and C have too much background staining of granular tubules, D has too much staining of large blood vessels? 

the F at the end of the title

Author Response

Reviewer 1

ITs difficult to get excited by this manuscript as these authors have already shown the lipoic acid effect on irradiation induced damage of salivary glands before.  Specifically they have also shown the AQP5 and the salivary secretion results before.  What is new and more interesting is the effects on nerves and although the GFRa2, neurturin and BDNF results are interesting what is missing is the functional tests of the nerves.  THis could be achieved in two ways -either electrically stimulating nerves or using taste to show functional integrity. Their acetyl cholinesterase staining is poor and does not show the nerves.  Thus we are left with an unfinished conclusion that lipoic acid probably preserves the nerves in the same way that the parenchyma is preserved.  The other results, looking at stem cell markers is too speculative to be useful.  qPCR may show message levels but nothing more and does not show regenerative potential.  I think most of these results should be removed. The writing is generally ok but there are lapses in understanding eg in discussion suggesting that parasympathetic stimulation increases apical intracellular calcium (which is does) which is then released into the ER and the ductal (which it does not).  

- We appreciate your critical comments.

- The utility of exogenous ALA in the treatment of diverse conditions, including diabetes, atherosclerosis, insulin resistance, neuropathy, neurodegenerative diseases, and ischemia-reperfusion injury, has been examined in clinical and experimental studies (Takaoka et al., 2002; Amudha et al., 2007; Alegre et al., 2010; Wongmekiat O et al., 2013). Most of studies are focused on antioxidant properties for ALA. Thus, we tried to find another ALA’s role in a previous study (Kim et al., 2016) besides antioxidant properties and that’s why the aim of a previous study was to assess the anti-inflammatory and fibrotic effects, as well as normal tissue injury and dysfunction, of ALA on the radiation-induced salivary glands injury.

- However, in our previous study (Kim et al., 2016), saliva is significantly reduced even if the salivary glands were partially intact after irradiation. Therefore, we hypothesized that other some factors can exist to control saliva secretion in irradiation conditions. In the current study, we tried to focus the effects of ALA on stem cells, soluble factors, and nerve innervation from out of salivary parenchymal cells.

- Interactions between salivary ductal epithelial cells and parasympathetic neuronal cells are required for epithelial regeneration in salivary tissue damage including irradiation as well as gland development (Knox et al., 2010 and 2013; Ferreira et al., 2018). In particular, neurturin (a neurotrophic factor) and its receptor GFRa2 is related to restoring parasympathetic function, which improved innervation and epithelial cells regeneration in radiation insult (Knox et al., 2013). These studies inform that targeting the epithelial-parasympathetic innervation is important to stimulate epithelial regeneration and salivary function in injury-induced salivary dysfunction.

- The salivary gland itself is one of organs with slow turnover activity and also harbors resident stem cell population. These stem cells play a crucial role for regenerating the parenchymal of the salivary gland. Several stem cell surface markers are well-established as CD24, CD29, CD49f, c-kit, and Sca-1 (Pringle et al., 2016).

- Thus, in the current study, to investigate the effects of ALA on regenerating parenchymal of the salivary gland in radiation-induced injury, we examined the expression levels of BDNF, neurturin, GFRa2, and stem cell markers.

- This study is a sort of preliminary one about regeneration-involved effects of ALA on the salivary hypofunction by irradiation. Authors agreed with keeping all of these data on the current manuscript, even your comments such as “most of these results should be removed”. As you mentioned, it is important to do intracellular Ca concentration, electric stimulation or taste function. We are already ongoing in further projects including your comments.

refernces)

Takaoka, M., Ohkita, M., Kobayashi, Y., Yuba, M. & Matsumura, Y. Protective effect of alpha-lipoic acid against ischaemic acute renal failure in rats. Clin Exp Pharmacol Physiol. 3, 189–194 (2002).

Amudha, G., Josephine, A., Sudhahar, V. & Varalakshmi, P. Protective effect of lipoic acid on oxidative and peroxidative damage in cyclosporine A-induced renal toxicity. Int Immunopharmacol. 7, 1442–1449 (2007).

Alegre, Vde. S., Barone, J. M., Yamasaki, S. C., Zambotti,Villela. L. & Silveira, P. F. Lipoic acid effects on renal function, aminopeptidase activities and oxidative stress in Crotalus durissus terrificus envenomation in mice. Toxicon. 56(3). 402-10 (2010).

Wongmekiat O, Leelarungrayub D, Thamprasert K. Alpha-lipoic acid attenuates renal injury in rats with obstructive nephropathy. Biomed Res Int.138719 (2013)

Kim JH, Kim KM, Jung MH, Jung JH, Kang KM, Jeong BK, Kim JP, Park JJ, Woo SH. Protective effects of alpha lipoic acid on radiation-induced salivary gland injury in rats. Oncotarget. 7(20):29143-29153 (2016)

Knox, S.M., Lombaert, I.M., Reed, X., Vitale-Cross, L., Gutkind, J.S., and Hoffman, M.P.. Parasympathetic innervation maintains epithelial progenitor cells during salivary organogenesis. Science 329, 1645–1647. (2010)

Knox, S.M., Lombaert, I.M., Haddox, C.L., Abrams, S.R., Cotrim, A., Wilson, A.J., and Hoffman, M.P. Parasympathetic stimulation improves epithelial organ regeneration. Nat. Commun. 4, 1494. (2013).

Joao N A Ferreira, Changyu Zheng , Isabelle M A Lombaert , Corinne M Goldsmith, Ana P Cotrim, Jennifer M Symonds, Vaishali N Patel, Matthew P Hoffman. Neurturin Gene Therapy Protects Parasympathetic Function to Prevent Irradiation-Induced Murine Salivary Gland Hypofunction. Mol Ther Methods Clin Dev, 9, 172-180 (2018).

SARAH PRINGLE, MARTTI MAIMETS, MARIANNE VAN DER ZWAAG, MONIQUE A. STOKMAN, DJOKE VAN GOSLIGA, ERIK ZWART, MAX J.H. WITJES, GERALD DE HAAN, RONALD VAN OS, ROB P. COPPES. Human Salivary Gland Stem Cells Functionally Restore Radiation Damaged Salivary Glands. STEM CELLS. 34:640–652 (2016)

Minor points: 

Hoffman is not a suitable reference

Fig 3 legend makes no sense. Fig 3a needs MW markers, B and C have too much background staining of granular tubules, D has too much staining of large blood vessels? 

- 1) The reference (Hoffman MP. 2013) is deleted. 2) Size markers are added to all figures with immunoblots and also descibed in figure legends. 3) To overcome background staining in figure 3B and C, each positive signal for AchE and neurofilaments were analyzed quantitatively by measuring signal intensity. 4) We regret to give a confusing description. Those are not large blood vessels. They are ducts in the salivary glands. We added to following sentence on the figure legend in the revised manuscript. “* indicate the ducts in SGs.”

Reviewer 2 Report

This is an experimental study that estimated how alpha lipoic acid protects salivary gland damage from irradiation. As the authors mentioned, this study may lead to the prevention of salivary gland damage when patients with head and neck cancer receive irradiation therapy. There was the previous paper from the same authors which reported ALA preserves acinoductal integrity and acinar cell secretary function following irradiation. And in the current study, the authors examined further mechanisms of the function of ALA, preserving aquaporin-5 expression, parasympathetic innervation, salivary tropic factor levels, and stem cell expression.

I am a head and neck oncologist and I really long for a possible treatment which prevents salivary gland degeneration while radiation therapy. This study seems to be really promising and I am waiting for the clinical application of this study. However, I have only one question which may jeopardise the clinical application of this study.

Major point

After reading this manuscript and the previous paper, I doubt that ALA prevents the oxidative damage of the tissue and reduces the effect of irradiation itself. If so, ALA may reduce anticancer effect of irradiation and we have to give up the clinical application of this material to the patients who receive radiation therapy for their head and neck cancer. Please make a comment or give us information from the previous papers about this issue, if possible.

Minor point

I don’t know the abbreviations like AQP5, GFRa2, AchE, BDNF may not have to be explained in the manuscript of the field of molecular science. From the view point of oncologist, it may helpful that the authors explain the formal names of these abbreviations such as, Aquaporin, Glial cell-line derived neurotrophic factor family receptor alpha 2, Acetylcholinesterase, Brain-derived neurotrophic factor and so on. I cannot understand the meaning of negative values of Figure 4B, Serum BDNF.

Author Response

Reviewer 2

This is an experimental study that estimated how alpha lipoic acid protects salivary gland damage from irradiation. As the authors mentioned, this study may lead to the prevention of salivary gland damage when patients with head and neck cancer receive irradiation therapy. There was the previous paper from the same authors which reported ALA preserves acinoductal integrity and acinar cell secretary function following irradiation. And in the current study, the authors examined further mechanisms of the function of ALA, preserving aquaporin-5 expression, parasympathetic innervation, salivary tropic factor levels, and stem cell expression. I am a head and neck oncologist and I really long for a possible treatment which prevents salivary gland degeneration while radiation therapy. This study seems to be really promising and I am waiting for the clinical application of this study. However, I have only one question which may jeopardise the clinical application of this study.

Major point

After reading this manuscript and the previous paper, I doubt that ALA prevents the oxidative damage of the tissue and reduces the effect of irradiation itself. If so, ALA may reduce anticancer effect of irradiation and we have to give up the clinical application of this material to the patients who receive radiation therapy for their head and neck cancer. Please make a comment or give us information from the previous papers about this issue, if possible.

- We appreciate your considerate comment. I am also head and neck surgical oncologist and consider also these topics, always. ALA-mediated in vitro or in vivo experiments in cancer cells were necessary to be added in order to better interpret. At the beginning of this study, this study was designed through a feasibility study about cancer cells growth and/or cancer cells death and ALA treatment. It has been reported that ALA can act as anti-cancer drug in various tumors (table, below). Thus, authors agree with skip studies on ALA effects in cancer cells in the current study.

Authors

Year

Cancer

ALA’s main functions

Farhat D et al

2020

breast cancer cells

abrogates IGF-1R maturation by inhibiting the CREB/furin axis

Yang Y et al

2019

Gastric Cancer Cells

Inhibits Proliferation and Invasion of Human Gastric Cancer Cells via Suppression of STAT3-Mediated MUC4 Gene Expression

Jeon MJ et al

2015

thyroid cancer cells

inhibits proliferation and epithelial mesenchymal transition

Kafara P et al

2015

ovarian carcinoma cells

decreases Mcl-1, Bcl-xL and up regulates Bim leading to cell death

Dörsam B and

Fahrer J

2015

novel class of anticancer agents targeting mitochondria

Dörsam B et al

2015

colorectal cancer cells

induces p53-independent cell death and potentiates the cytotoxicity of 5-fluorouracil

Yamasaki M et al

2014

bladder cancer cells

suppresses migration and invasion via downregulation of cell surface β1-integrin expression

Michikoshi H et al

2013

non-small cell lung cancer cells

inhibition of proliferation and met phosphorylation

Feuerecker B et al

2012

tumor cells

inhibits cell proliferation of tumor cells in vitro and in vivo

And also, the ALA has an anti-cancer effect at high levels, and doesn’t affect at low level in the head and neck cancer cell lines (not published). We are also going on the new project including 1) anti-cancer function of ALA and 2) protective function for normal tissue of ALA on the mice model with tumors and radiation. Thus, we think ALA will be a good option for prevention radiation complications even if the road is too long.

Minor point

I don’t know the abbreviations like AQP5, GFRa2, AchE, BDNF may not have to be explained in the manuscript of the field of molecular science. From the view point of oncologist, it may helpful that the authors explain the formal names of these abbreviations such as, Aquaporin, Glial cell-line derived neurotrophic factor family receptor alpha 2, Acetylcholinesterase, Brain-derived neurotrophic factor and so on. I cannot understand the meaning of negative values of Figure 4B, Serum BDNF.

- We apologize for poor description and wrong analysis. Each word (AQP5, GFRa2, AchE, BDNF) was described to formal names at the first sentence where those words in the revised manuscript. We reanalyzed ELISA data for serum BDNF. Corrected data is added to figure 4B in the revised manuscript.

Reviewer 3 Report

The manuscript by Kim et al. reports on the activity of alpha lipoic acid (ALA) against salivary glands in a rat model. The study is interesting but several issues warrant further attention by the authors.

  1. Introduction: The last paragraph describing ALA is insufficient. The authors should include a better description of the rationale behind exploring ALA and comment on how the present study builds on their previous work.  Was there a hypothesis? If so, the rationale for the hypothesis should be stated.
  2. On a related note, the authors should avoid vague sentences “ALA has demonstrated efficacy in preventing various pathological processes”.
  3. Figure 1A. It appears ALA did not have any effect on the body weight of animals suggestive of a local effect. It might also be easier to plot the change in body weights as curves rather than bar graphs. The authors should include a line in the discussion. Was there any evidence of radiation dermatitis in the animals? If so, please describe the findings.
  4. Figure 1B. Why are the salivary gland weights of control animals (not exposed to irradiation) decreasing over the course of the study?
  5. Figure 1C: Saliva secretion measurements for week 6 and 8 are missing.
  6. Figure 2A: The effect of ALA alone on AQP5 is presented in Supp. Fig 1 and is shown as an absolute value for the expression while AQP5 levels in Figure 2 are shown as a relative change. Unclear what fold change refers to? Beta actin normalization? Include the methods for quantifying western blot analysis and plot all groups on the same graph with either the relative change or expression levels reported consistently.
  7. Figure 2B: The photomicrographs of AQP5 stained sections are of insufficient resolution and do not convincingly validate the immunoblotting results. Please include higher resolution images with scale bars for the figure and report quantitative or semi-quantitative results of AQP5 staining.
  8. Figure 3. Clarify fold change similar to Fig. 2. IHC data is reasonably convincing but higher magnification images and quantitation would be beneficial for the readers. If possible, provide images of matching fields.
  9. Figure 4. 6 week data is notable missing.
  10. Figure 6. The Sca-1 expression data is not convincing. Is there supportive IHC data?
  11. Discussion: Although the authors provide a reasonable discussion of their findings, differences in the kinetics between the changes in the neuronal markers (week 6 vs. week 8 vs. week 12) should be discussed in the context of the histologic and functional changes (saliva secretion) associated with radiation damage and recovery.
  12. Discussion: A discussion of clinical implications of their findings (potential ALA for clinical use; safety?) along with a paragraph on limitations of their study should be included.

Minor comments:

  1. Were 3 rats from the total of 12 rats per cohort were euthanized at the 4 time points? The sample sizes for the individual assays should be included in the figure legends or at least in the description of individual results. Unclear how many samples were used for immunoblotting (presumably 3) and how many samples/fields were used for immunostaining.

Author Response

Reviewer 3

The manuscript by Kim et al. reports on the activity of alpha lipoic acid (ALA) against salivary glands in a rat model. The study is interesting but several issues warrant further attention by the authors.

  1. Introduction: The last paragraph describing ALA is insufficient. The authors should include a better description of the rationale behind exploring ALA and comment on how the present study builds on their previous work.  Was there a hypothesis? If so, the rationale for the hypothesis should be stated.

- We appreciate your critical comments.

- The utility of exogenous ALA in the treatment of diverse conditions, including diabetes, atherosclerosis, insulin resistance, neuropathy, neurodegenerative diseases, and ischemia-reperfusion injury, has been examined in clinical and experimental studies (Takaoka et al., 2002; Amudha et al., 2007; Alegre et al., 2010; Wongmekiat O et al., 2013). Most of studies are focused on antioxidant properties for ALA. Thus, we tried to find another ALA’s role in a previous study (Kim et al., 2016) besides antioxidant properties and that’s why the aim of a previous study was to assess the anti-inflammatory and fibrotic effects, as well as normal tissue injury and dysfunction, of ALA on the radiation-induced salivary glands injury.

- However, in our previous study (Kim et al., 2016), saliva is significantly reduced even if the salivary glands were partially intact after irradiation. Therefore, we hypothesized that other some factors can exist to control saliva secretion in irradiation conditions. In the current study, we tried to focus the effects of ALA on stem cells, soluble factors, and nerve innervation from out of salivary parenchymal cells.

- Interactions between salivary ductal epithelial cells and parasympathetic neuronal cells are required for epithelial regeneration in salivary tissue damage including irradiation as well as gland development (Knox et al., 2010 and 2013; Ferreira et al., 2018). In particular, neurturin (a neurotrophic factor) and its receptor GFRa2 is related to restoring parasympathetic function, which improved innervation and epithelial cells regeneration in radiation insult (Knox et al., 2013). These studies inform that targeting the epithelial-parasympathetic innervation is important to stimulate epithelial regeneration and salivary function in injury-induced salivary dysfunction.

- The salivary gland itself is one of organs with slow turnover activity and also harbors resident stem cell population. These stem cells play a crucial role for regenerating the parenchymal of the salivary gland. Several stem cell surface markers are well-established as CD24, CD29, CD49f, c-kit, and Sca-1 (Pringle et al., 2016).

- Thus, in the current study, to investigate the effects of ALA on regenerating parenchymal of the salivary gland in radiation-induced injury, we examined the expression levels of BDNF, neurturin, GFRa2, and stem cell markers.

- This study is the first story about regeneration-involved effects of ALA on the salivary hypofunction by irradiation.

refernces)

Takaoka, M., Ohkita, M., Kobayashi, Y., Yuba, M. & Matsumura, Y. Protective effect of alpha-lipoic acid against ischaemic acute renal failure in rats. Clin Exp Pharmacol Physiol. 3, 189–194 (2002).

Amudha, G., Josephine, A., Sudhahar, V. & Varalakshmi, P. Protective effect of lipoic acid on oxidative and peroxidative damage in cyclosporine A-induced renal toxicity. Int Immunopharmacol. 7, 1442–1449 (2007).

Alegre, Vde. S., Barone, J. M., Yamasaki, S. C., Zambotti,Villela. L. & Silveira, P. F. Lipoic acid effects on renal function, aminopeptidase activities and oxidative stress in Crotalus durissus terrificus envenomation in mice. Toxicon. 56(3). 402-10 (2010).

Wongmekiat O, Leelarungrayub D, Thamprasert K. Alpha-lipoic acid attenuates renal injury in rats with obstructive nephropathy. Biomed Res Int.138719 (2013)

Kim JH, Kim KM, Jung MH, Jung JH, Kang KM, Jeong BK, Kim JP, Park JJ, Woo SH. Protective effects of alpha lipoic acid on radiation-induced salivary gland injury in rats. Oncotarget. 7(20):29143-29153 (2016)

Knox, S.M., Lombaert, I.M., Reed, X., Vitale-Cross, L., Gutkind, J.S., and Hoffman, M.P.. Parasympathetic innervation maintains epithelial progenitor cells during salivary organogenesis. Science 329, 1645–1647. (2010)

Knox, S.M., Lombaert, I.M., Haddox, C.L., Abrams, S.R., Cotrim, A., Wilson, A.J., and Hoffman, M.P. Parasympathetic stimulation improves epithelial organ regeneration. Nat. Commun. 4, 1494. (2013).

Joao N A Ferreira, Changyu Zheng , Isabelle M A Lombaert , Corinne M Goldsmith, Ana P Cotrim, Jennifer M Symonds, Vaishali N Patel, Matthew P Hoffman. Neurturin Gene Therapy Protects Parasympathetic Function to Prevent Irradiation-Induced Murine Salivary Gland Hypofunction. Mol Ther Methods Clin Dev, 9, 172-180 (2018).

SARAH PRINGLE, MARTTI MAIMETS, MARIANNE VAN DER ZWAAG, MONIQUE A. STOKMAN, DJOKE VAN GOSLIGA, ERIK ZWART, MAX J.H. WITJES, GERALD DE HAAN, RONALD VAN OS, ROB P. COPPES. Human Salivary Gland Stem Cells Functionally Restore Radiation Damaged Salivary Glands. STEM CELLS. 34:640–652 (2016)

  1. On a related note, the authors should avoid vague sentences “ALA has demonstrated efficacy in preventing various pathological processes”.

- We regret to show this sentence. This sentence is deleted in the revised manuscript

  1. Figure 1A. It appears ALA did not have any effect on the body weight of animals suggestive of a local effect. It might also be easier to plot the change in body weights as curves rather than bar graphs. The authors should include a line in the discussion. Was there any evidence of radiation dermatitis in the animals? If so, please describe the findings.

- 1) Frankly, it is so hard if ALA has not systemic effect but local effect from figure 1A. But, figure1B shows that weight of SGs is ameliorated by ALA treatment in some time points. So, we think ALA’s effect could be faced to a local effect but not sure about that. 2) Plots for body and salivary gland weight are changed to curve one and are added to figure 1 in the revised manuscript. 3) Actually we did not find the any dermatitis on the skin from all rats. But, there is an evidence for radiation dermatitis as following.

Experiments with fractionated x-ray treatment of the skin of pigs: 1. Fractionation up to 28 days. Br J Radiol. 1963;36:188–196

The effect of divided doses of 15 MeV electrons on the skin response of mice. Int J Radiat Biol. 1965;9:241–252

  1. Figure 1B. Why are the salivary gland weights of control animals (not exposed to irradiation) decreasing over the course of the study?

- Frankly, we can’t explain exactly, it is just result itself and also may be a natural process in aging. There is evidence that age is related to functional or morphological changes of salivary glands (Choi et al., 2013).

Reference)

Choi JS, Park IS, Kim SK, Lim JY, Kim YM. Analysis of age-related changes in the functional morphologies of salivary glands in mice. Arch Oral Biol. 2013 Nov;58(11):1635-42.

  1. Figure 1C: Saliva secretion measurements for week 6 and 8 are missing.

- Yes, we did not measure saliva secretion at those time points, intentionally. To measure saliva, pilocarpine is used. Pilocarpine is known to have serious side effects including mortality. This study aims to long-term effects of ALA. So, mortality is critical in this study. Therefore, we decided to examine saliva secretion in the beginning and last. Please, forgive to our limitations.

  1. Figure 2A: The effect of ALA alone on AQP5 is presented in Supp. Fig 1 and is shown as an absolute value for the expression while AQP5 levels in Figure 2 are shown as a relative change. Unclear -what fold change refers to? Beta actin normalization? Include the methods for quantifying western blot analysis and plot all groups on the same graph with either the relative change or expression levels reported consistently.

- We totally agree with your comment. All data values are unified as fold changes. The fold change is calculated as the ratio of the final value in the each group to the value in control group at 2 week (set as “1”). In immunoblots and PCR analysis, the expression levels for each target are normalized to b-actin and GAPDH, respectively. Graph with quantitative values in supplementary figure 1 is changed to fold changes in the revised manuscript.

  1. Figure 2B: The photomicrographs of AQP5 stained sections are of insufficient resolution and do not convincingly validate the immunoblotting results. Please include higher resolution images with scale bars for the figure and report quantitative or semi-quantitative results of AQP5 staining.

- We apologize for poor description. Scale bar is added to all immunostaining images and figure legends as well as figure 2. AQP-5-positive signals were in quantitative analysis by signal intensity and the images were changed with higher resolution. The data is added to the revised manuscript.

  1. Figure 3. Clarify fold change similar to Fig. 2. IHC data is reasonably convincing but higher magnification images and quantitation would be beneficial for the readers. If possible, provide images of matching fields.

- All IHC data as well as figure 3 were analyzed by signal intensity in 10 random fields. This was added to figure 3D in the revised manuscript. IHC data for GFRa2 is deleted, intentionally. It is due to overlap immunoblot data in figure 3A.

  1. Figure 4. 6 week data is notable missing.

- We apologize for wrong upload. We realized that after submission done. Corrected data is added to figure 4B in the revised manuscript.

  1. Figure 6. The Sca-1 expression data is not convincing. Is there supportive IHC data?

- We totally agree with your comment. To confirm effect of ALA on the preservation of stem cell population, we performed immunostaining for c-Kit. c-Kit is also known to SG stem/progenitor cell marker in mice and rodent (Nanduri et al., 2011; Pringle et al., 2016). The c-Kit-positive signals were detected in the excretory ductal cells of the submandibular gland (arrow in Con and ALA). Localization of c-Kit-positive signals was well coincided with the previous report (Nanduri et al., 2011). Irradiation significantly downregulated c-Kit expression particularly in ductal structures in all time points (RT), whereas ALA treatment prior to irradiation restored c-Kit expression significantly in 2 weeks but tended to ameliorate in other time points (ALA+RT). This data was added to figure 6B and D in the revised manuscript.

Reference)

Lalitha S Y Nanduri, Martti Maimets, Sarah A Pringle et al. Regeneration of Irradiated Salivary Glands With Stem Cell Marker Expressing Cells. Radiother Oncol. 2011;99(3): 367-72.

Pringle S, Maimets M, van der Zwaag M et al. Human Salivary Gland Stem Cells Functionally Restore Radiation Damaged Salivary Glands. Stem Cells 2016; 34: 640-652.

  1. Discussion: Although the authors provide a reasonable discussion of their findings, differences in the kinetics between the changes in the neuronal markers (week 6 vs. week 8 vs. week 12) should be discussed in the context of the histologic and functional changes (saliva secretion) associated with radiation damage and recovery.

- Although radiation itself destroys the salivary gland, sometimes the salivary cells are alive.

However, even the cells are intact, the saliva secretin was not enough than we expect. This suggests that radiation-induced saliva hypofunction is not limited to SG tissue itself and also continues for long time. Thus, we think that there is a shift to change the focus from SG tissue restoration to extra-SG tissue regeneration. It has been reported that epithelial (SGs itself)-parasympathetic (extra-SGs) innervation is required for the saliva secretion. In the current data, figure 3, 4, and 5 shows parasympathetic innervation is crucial for keeping the salivary function and ALA can ameliorate salivary dysfunction by irradiation through preservation of parasympathetic nerve. But, we should do to confirm ALA’s role on epithelial-parasympathetic innervation in radiation-induced SGs injury in the future study.

  1. Discussion: A discussion of clinical implications of their findings (potential ALA for clinical use; safety?) along with a paragraph on limitations of their study should be included.

-We appreciate your considerate comments. As mentioned in your comments #1, ALA is in clinic use and is mainly used for improving symptoms of the patients with diabetic neuropathy. But, its use and effects are still controversial in other clinic conditions. Although this study doesn’t show effects of ALA on head and neck cancer, we believe ALA is capable of treating radiation-induced neuropathy. In conclusion, the current data provides that ALA has a promising therapeutic potential against radiation-induced salivary dysfunction.

Minor comments:

  1. Were 3 rats from the total of 12 rats per cohort were euthanized at the 4 time points? The sample sizes for the individual assays should be included in the figure legends or at least in the description of individual results. Unclear how many samples were used for immunoblotting (presumably 3) and how many samples/fields were used for immunostaining?

- We apologize for poor description. Number of animals and samples in all figures including immunoblotting and immunostaining was described (yellow-highlighted) on all figure legends in revised manuscript.